# Screening of a Fraction with Higher Amyloid β Aggregation Inhibitory Activity from a Library Containing 210 Mushroom Extracts Using a Microliter-Scale High-Throughput Screening System with Quantum Dot Imaging

**DOI:** 10.3390/foods13233740

**Published:** 2024-11-22

**Authors:** Gegentuya Huanood, Mahadeva M. M. Swamy, Rina Sasaki, Keiya Shimamori, Masahiro Kuragano, Enkhmaa Enkhbat, Yoshiko Suga, Masaki Anetai, Kenji Monde, Kiyotaka Tokuraku

**Affiliations:** 1Graduate School of Engineering, Muroran Institute of Technology, Muroran 050-8585, Japan; gegentuya1996@gmail.com (G.H.); sasaki-rin@awi.co.jp (R.S.); simauma8476@gmail.com (K.S.); gano@muroran-it.ac.jp (M.K.); 2Faculty of Advanced Life Science, Hokkaido University, Sapporo 001-0021, Japan; mswamy.madegowda@gmail.com (M.M.M.S.); enkhmaae@mas.ac.mn (E.E.); suga@sci.hokudai.ac.jp (Y.S.); glycinoeclepin-a@hb.tp1.jp (M.A.); kmonde@sci.hokudai.ac.jp (K.M.)

**Keywords:** mushroom, quantum dot nanoprobes, amyloid β, Alzheimer’s disease, inhibitor, natural compounds

## Abstract

Alzheimer’s disease (AD) is a highly prevalent neurodegenerative disease hallmarked by amyloid plaques and neurofibrillary tangles. Amyloid plaques are formed by the amyloid β (Aβ) aggregation, so substances that inhibit this aggregation are useful for preventing and treating AD. Mushrooms are widely used medicinal fungi with high edible and nutritional value. Mushrooms have a variety of biologically active ingredients, and studies have shown that they have certain effects in anti-bacterial, anti-oxidation, anti-inflammatory, anti-tumor, and immune regulation. Previously, we developed a microliter-scale high-throughput screening (MSHTS) system using quantum dot (QD) nanoprobes to screen Aβ aggregation inhibitors. In this study, we appraised the Aβ aggregation inhibitory activity of 210 natural mushrooms from Hokkaido (Japan) and found 11 samples with high activity. We then selected *Elfvingia applanata* and *Fuscoporia obliqua* for extraction and purification as these samples were able to suppress Aβ-induced neurocytotoxicity and were readily available in large quantities. We found that the ethyl acetate (EtOAc) extract of *E. applanata* has high Aβ aggregation inhibitory activity, so we performed silica gel column chromatography fractionation and found that fraction 5 (f5) of the EtOAc extract displayed the highest Aβ aggregation inhibitory activity among all mushroom samples. The half-maximal effective concentration (EC_50_) value was 2.30 µg/mL, higher than the EC_50_ of 10.7 µg/mL for rosmarinic acid, a well-known Aβ aggregation inhibitor. This inhibitory activity decreased with further purification, suggesting that some compounds act synergistically. The f5 fraction also inhibited the deposition of Aβ aggregates on the cell surface of human neuroblastoma SH-SY5Y cells. Our expectation is that f5, with additional tests, may eventually prove to be an inhibitor for the prevention of AD.

## 1. Introduction

The number of patients with dementia has increased rapidly in recent years as the population ages [1]. Alzheimer’s disease (AD) is a degenerative disease of the central nervous system that occurs frequently in the elderly and pre-elderly population, characterized by progressive cognitive and behavioral impairment, and is the most common type of dementia, accounting for 60–70% of dementia patients [2,3,4]. According to the amyloid hypothesis, the main cause of AD is the abnormal accumulation of amyloid β (Aβ) in the brain. Aβ is a short peptide produced by the cleavage of amyloid precursor protein, which easily aggregates to form insoluble fibers under pathological conditions, thereby forming plaques in the brain. The accumulation of Aβ and the formation of plaques will trigger neurotoxicity, ultimately leading to neuronal death and cognitive decline [1]. Currently, some progress has been made in immunotherapy drugs that target Aβ. The United States Food and Drug Administration approved the Aβ-targeting antibody drugs Aducanumab on 21 June 2021 and Lecanemab on 22 January 2023 [5]. Although these drugs have achieved certain results in clinical applications, they are still unable to effectively cure AD, and both display certain side effects [6]. Aducanumab is likely to increase the incidence of amyloid-related imaging abnormalities in the case of cerebrospinal effusion or hemorrhages, indicating that it has potential risks [7]. Some studies have also suggested that Lecanemab does not slow down the cognitive decline of AD patients [8]. Therefore, these two AD treatment drugs are still controversial. Considering the side effects and high costs of these drugs, it is unclear which patients might benefit from them [9]. In recent years, researchers from various countries have made considerable progress in screening Aβ aggregation inhibitors, with natural extracts increasingly showing strong Aβ aggregation inhibitory activity [10]. Therefore, the prevention of AD may be more effective if functional foods are consumed as part of a diet rather than following a treatment path.

Mushrooms are large fungi that belong to the phylum Basidiomycetes or Ascomycetes [11]. More than 140,000–160,000 species of mushrooms have been discovered so far, making them an ideal natural multifunctional food. Most mushrooms are edible, and some species are not only high in nutritional value but also have high medicinal value, such as shiitake mushrooms, *Agaricus bisporus*, and *Ganoderma lucidum* [12,13]. Currently, researchers have identified and purified a variety of bioactive substances from mushrooms, such as digestible proteins, carbohydrates, and terpenes [14]. These bioactive ingredients not only have antioxidant, anti-bacterial, and anti-tumor effects but also show potential neuroprotective properties [15].

Oral administration of *Hericium erinaceus* was shown to significantly increase cell viability and reduce the release of lactate dehydrogenase, thereby reducing Aβ25-35-triggered damage in PC12 cells [16]. Ergosterol extracted from mushrooms (*Auricularia polytricha*, *Lignosus rhinocerus*, *Ganoderma lucidum*, and *Agaricus bisporus*) inhibited the production of Aβ by promoting neurite growth of neuroblastoma cells and also inhibited the activity of β-secretase and γ-secretase by regulating the Ten-4/ERK/CREB/GAP-43 signaling pathway [17]. Therefore, edible mushrooms with neuroprotective properties have attracted attention as a possible countermeasure for AD.

Currently, thioflavin T (ThT) and transmission electron microscopy (TEM) are widely used to evaluate the Aβ aggregation inhibitory activity of various substances and to observe Aβ aggregation [18]. ThT experiment is a commonly used fluorescence analysis method. During the formation of Aβ fibrils, ThT can bind to its β-sheet and cause an increase in fluorescence intensity. By measuring the fluorescence intensity of ThT, the amount of Aβ aggregation can be quantified [19]. However, ThT dyes and fluorescent substances may cause internal filtering effects, and inhibitors may compete with ThT for binding to Aβ [20]. The principal disadvantage of TEM is its limited quantitative capability because of the complexity of sample preparation, including drying and staining. Recently, we reported a real-time imaging method for Aβ aggregation with quantum dot (QD) and developed a microliter-scale high-throughput screening (MSHTS) system using the QD imaging technology [21,22]. The advantage of the MSHTS system is that high-throughput screening can be performed using 1536-well plates with a sample volume of only 5 µL.

QD can be functionalized with a variety of biomolecules, which allows them to selectively bind to specific biomolecular targets, such as Aβ aggregates. This customization enhances specificity and sensitivity, especially in complex environments like biological tissues. QD has relatively long fluorescence lifetimes compared to organic dyes [23]. This property can be beneficial in time-resolved fluorescence imaging, allowing for background signal reduction and better discrimination in complex samples. Recently, some researchers developed highly homogeneous and dispersed AgBiS2 QDs, delivering a champion power conversion efficiency of approximately 8% in the QD solar cells with outstanding shelf-life stability [24]. A novel nanosensor based on graphene QD was developed by some researchers to detect azodicarbonamide in foods [25].

In this study, the purpose was to evaluate the Aβ aggregation inhibitory activity of different species of mushrooms collected in Hokkaido, Japan. There are about 1000 named species of mushrooms in Hokkaido, and the 210 species used in this experiment (excluding slime molds and cordyceps) account for about 20% of them. These 210 mushrooms include two phyla, six classes, and 17 orders (Appendix A). We used this MSHTS system based on QD fluorescence imaging to rapidly and accurately evaluate the Aβ aggregation inhibitory activity of 210 mushrooms. We found that 11 mushrooms showed high Aβ aggregation inhibitory activities. With further purification, the ethyl acetate (EtOAc) extract fraction 5 (f5) of *Elfvingia applanata* (t037) showed the highest inhibitory activity. In summary, we hope that f5 can be developed into an effective inhibitor for preventing AD in the future. At the same time, the results of this study also show that the MSHTS system will become an important method for evaluating amyloid aggregation inhibition.

## 2. Materials and Methods

### 2.1. Materials

Human Aβ_42_ (4349-v, Peptide Institute Inc., Osaka, Japan) and Cys-conjugated Aβ_40_ (23519, Anaspec Inc., Fremont, CA, USA) were procured from commercial suppliers. Sodium dodecyl sulfate (SDS) was purchased from FUJIFILM Wako Pure Chemical Corp. (Osaka, Japan). Thiazolyl Blue Tetrazolium Bromide (MTT) for MTT assay and Poly-D-Lysine (PDL) used in cell culture were procured from Sigma-Aldrich (St. Louis, MO, USA). Nerve growth factor (NGF) induced neuronal differentiation was obtained from Cosmo Bio (Tokyo, Japan).

### 2.2. Preparation of the QDAβ Nanoprobe

The QDAβ was prepared using QD-PEG-NH_2_ (Qdot^TM^ 605 ITK^TM^ Amino (PEG) Quantum dot; Q21501MP, Thermo Fisher Scientific, Waltham, MA, USA) and Cys-conjugated Aβ_40_ according to our previous report [26]. The concentration of QDAβ was determined by comparing absorbance at 350 nm of the prepared QDAβ and unlabeled QD-PEG-NH_2_.

### 2.3. The MSHTS System

The half-maximal effective concentration (EC_50_) values of all mushroom extracts were calculated by a modified automated MSHTS system according to our previous report [26]. Particularly, various concentrations of mushroom extracts, 25 nM QDAβ, and 25 μM Aβ_42_ were prepared and incubated at 37 °C for 24 h in a 1536-well plate (782096, Greiner, Kremsmünster, Austria). The images of each well were observed by an inverted fluorescence microscope (ECLIPSE Ti-E, Nikon, Tokyo, Japan) equipped with a color CMOS camera (DS-Ri2, Nikon). QD was imaged using a 4× objective lens (MRD00045, Nikon) and a TRITC filter set (TRITC-A-Basic-NTE, Semrock, NY, USA). The standard deviation (SD) value of the fluorescence intensity in the central area (432 × 432 pixels) of each well was measured using ImageJ software version 1.53b (National Institutes of Health, Bethesda, MD, USA). EC_50_ was calculated by global fit (Asymmetric sigmoidal, 5 parameter logistic) using Prism software (6.01, GraphPad software, San Diego, CA, USA).

### 2.4. Preparation of 210 Mushroom Crude Extracts

The 210 mushrooms from Hokkaido, Japan were harvested and cleaned of soil and other debris. They were then frozen and stored for use. Fresh mushroom samples contain approximately 90% water, so a 50 g fresh sample contains approximately 45 g (45 mL) of water. In this study, 50 g of mushroom sample was treated with 105 mL of acetone, resulting in extraction with 150 mL of 70% acetone. For extraction, acetone was added to roughly chopped mushrooms, left overnight, filtered with a funnel, and the solvent was completely removed using a rotary evaporator (N-1110, Eyela, Tokyo Rikakikai Co., Ltd., Tokyo, Japan). The obtained extracts were stored at −20 °C. The crude extract of each mushroom was dissolved in dimethyl sulfoxide (DMSO) (FUJIFILM Wako Pure Chemical Corp., Osaka, Japan) to prepare an extract solution with a concentration of 10 mg/mL. For *E. applanata,* in addition to those collected from Hokkaido, those purchased from Tochimoto Tenkaido Co., Ltd. (Osaka, Japan) were also used.

### 2.5. MTT Assay

Rat adrenal pheochromocytoma, PC12 cells, which were used as an in vitro model, were purchased from JCRB Cell Bank (Osaka, Japan). PC12 cells were added to Dulbecco’s Modified Eagle Medium (DMEM) containing 10% fetal bovine serum (FBS), 100 U/mL penicillin, and 100 μg/mL streptomycin, and the cells were cultured in a 5% CO_2_ incubator at 37 °C [27]. PC12 cells were prepared to a concentration of 2.5 × 10^4^ cells per well and seeded into PDL-coated 96-well plates (3860-096, AGC Techno Glass, Shizuoka, Japan) and incubated. After incubation for 24 h, the cells in DMEM were treated with 50 ng/mL NGF for 24 h to induce neuronal differentiation [27]. Subsequently, the cells were treated with 25 nM QDAβ, 25 μM Aβ_42_, and each of 11 mushroom extracts (50 μg/mL) and incubated for another 24 h. Then, the cells were treated with 5 mg/mL MTT and incubated for 4 h, the supernatant of each well was removed, and the formazan crystals were dissolved in 10% SDS/0.01 M HCl solution. After overnight incubation, the absorbance intensity (570 nm) was measured in a 96-well plate (3860-096, Iwaki AGC Techno Glass Co., Ltd., Shizuoka, Japan). Cell viability was evaluated as a percentage relative to DMSO-treated cells.

### 2.6. Solvent Partition and Fractionation of Elfvingia applanata (t037) and Fuscoporia obliqua (t100)

Dried *Elfvingia applanta* (t037) was cut into small pieces and ground/powdered properly using a blender (861-66L, Osterizer 16 speed blender, Wilmington, DE, USA). A mass (250 g) of t037 was soaked in 500 mL of methanol (MeOH) (99.8%, 25183-80, Kanto Chemical Co., Inc., Tokyo, Japan) at room temperature. After 24 h, MeOH was collected by vacuum filtration, and the solvent was evaporated under reduced pressure. The MeOH extraction procedure was repeated two more times after 24 h each. The combined MeOH extract was concentrated under reduced pressure to give a dark brown residue. The crude MeOH extract dissolved in 10% MeOH in water (300 mL) and partitioned with hexane (*n*-hexane) (200 mL × 3), diethyl ether (Et_2_O) (200 mL × 3), and EtOAc (200 mL × 3). The residue of each solvent partition was subjected to MHSTS assay. We found that the extract of EtOAc showed more Aβ aggregation inhibition activity compared to other solvent partitions. The EtOAc fraction was sub-fractionated into five fractions 1–5 (f1–f5) by silica gel flash column chromatography using chloroform (CHCl_3_) and MeOH as eluting solvents. The volume of each fraction and sub-fractionation was approximately 50 mL, and each fraction was determined by TLC results. The most active fraction, f5, was further fractionated, but inhibitory activity decreased. A similar extraction procedure was also used on *Fuscoporia obliqua* (t100). All dried extracts and fractions were dissolved in DMSO to a concentration of 10 mg/mL for use.

### 2.7. ThT Assay

A previously published ThT assay protocol was used, with some modifications to the experimental conditions [28]. To verify the MSHTS results of fraction 5 (f5) and fraction 3 of f5 (f5f3), we further evaluated the inhibitory activity of f5 and f5f3 on Aβ aggregation using ThT assay. The ThT assay was performed in a clear bottom polystyrene 384-well plate (3540, Corning, NY, USA). Aβ, rosmarinic acid (RA), and various concentrations of f5 and f5f3 were mixed with an equal volume of 40 μM ThT solution. The final concentrations were 1× PBS, 5% DMSO, 25 μM Aβ, 300 μM RA, 0.2, 2, and 20 μg/mL f5, 0.2, 2, and 20 μg/mL f5f3, and 20 μM ThT. Since Aβ aggregation was fully saturated by 24 h [21,22], ThT fluorescence intensity was measured in real time for 24 h using a microplate reader (SH-9000, Yamato, Tokyo, Japan).

### 2.8. Evaluation of the Inhibitory Effect of Mushroom Fractions on Aβ Deposition on the Surface of SH-SY5Y Cells

SH-SY5Y cells (EC94030304-F0), human neuroblastoma cells, were purchased from KAC (Kyoto, Japan). SH-SY5Y cells were added to a DMEM medium containing 10% FBS, 100 U/mL penicillin, and 100 μg/mL streptomycin, and the cells were cultured in a 5% CO_2_ incubator (MG-70C, TAITEC, Saitama, Japan) at 37 °C. SH-SY5Y cells were prepared to a concentration of 6.0 × 10^4^ cells and seeded into 10 μg/mL fibronectin-coated 96-well plates (5866-096, Iwaki AGC Techno Glass Co., Ltd., Shizuoka, Japan) and incubated for 24 h. Then, the medium was removed, and the cells were treated with 25 μM Aβ containing QDAβ and various concentrations of f5 and f5f3 and incubated for 24 h. Washed with 1× PBS (100 μL per well) once, then added 1× PBS (100 μL per well). To each solution, 4% paraformaldehyde phosphate buffer solution (163-20145, FUJIFILM Wako Pure Chemical Corp., Osaka, Japan) was added. After incubation for 20 min, images were observed by an inverted fluorescence microscope. The mean gray value (representing Aβ aggregation) in five different areas (each area: 1608 × 1608 pixels) of each well was measured using ImageJ software version 1.53b.

### 2.9. Statistical Analysis

All statistical analyses were conducted using Microsoft Excel (version 16; Redmond, WA, USA). A two-tailed Welch’s *t*-test was employed to evaluate differences between groups. The evaluation of EC_50_ was performed using linear regression with least squares in Excel. A *p*-value of less than 0.05 was considered statistically significant, and the following symbols were used to denote the level of significance: * (*p* < 0.05), ** (*p* < 0.01), and *** (*p* < 0.005).

## 3. Results

### 3.1. Evaluation of the Aβ Aggregation Inhibitory Activity of 210 Mushroom Species

An automated MSHTS system was used to assess the Aβ aggregation inhibitory activity of crude extracts of 210 mushroom species (Appendix A) collected in Hokkaido, Japan. Based on the evaluation results, 11 mushrooms with inhibitory activity were selected from among the 210 mushrooms (Figure 1). At an extract concentration of 100 µg/mL, Aβ aggregation was barely visible, but at lower concentrations, Aβ aggregation was clearly observed (Figure 1A). In our previous study, we evaluated the Aβ aggregation inhibitory activity of more than 500 plants, of which more than 50 plants, accounting for about 10% of the total, had EC_50_ values below 0.05 mg/mL (50 µg/mL), so we decided to use a value below 50 µg/mL as the high activity standard [26]. According to the EC_50_ values, the values of 11 mushroom extracts were less than 50 µg/mL, which showed high Aβ aggregation inhibitory activity (Figure 1B).

### 3.2. Effects of Mushroom Extracts on Aβ-Induced Neurocytotoxicity

PC12 cell viability is a relevant indicator for determining Aβ aggregation [29]. The researchers state that NGF treatment leads to the differentiation of PC12 cells and the relative number of differentiated cells in each case has been quantified [29]. We tested the 11 mushroom extracts with Aβ inhibitory activity, using the MTT assay, to verify whether they could inhibit Aβ-induced neurotoxicity in NGF-differentiated PC12 cells (Figure 2). The PC12 cell viability of the mixture of 25 μM Aβ_42_ and 50 μg/mL t208 mushroom extract was 40.53%, indicating that cell viability was not restored. The t100 and t020 mushroom extracts significantly increased PC12 cell viability at 50 μg/mL, increasing it to 59.98% and 65.26%, respectively, indicating that they had the best inhibitory effect on Aβ-induced neurotoxicity in PC12 cells. In contrast, the t039 and t034 extracts at 50 μg/mL reduced cell viability by about 4% compared with the Aβ_42_ group. Some mushroom extracts presumably contain cytotoxic substances, and even if they have high Aβ aggregation inhibitory activity, cell viability may not necessarily be high. Therefore, in this study, which also considers functional food applications, we focused on mushroom extracts with high Aβ aggregation inhibitory activity and cell viability enhancing activity.

### 3.3. Evaluation of the Inhibitory Activity of Different Solvents of Mushroom Extracts on Aβ Aggregation

t037 and t100 were selected for extraction and purification, as these samples were shown to suppress Aβ-induced neurocytotoxicity in rat PC12 cells and were readily available in large quantities (Table 1). In order to purify the active ingredients in the extracts, these two crude extracts were dissolved in five different solvents to re-evaluate Aβ aggregation inhibitory activity. The results showed that after 24 h of incubation, in the Et_2_O and EtOAc extracts of t037 at a concentration of 100 μg/mL, almost no aggregates were observed (Figure 3A). Similarly, only a small number of aggregates were observed in the Et_2_O and EtOAc extracts of t100 at a concentration of 100 μg/mL (Figure 3B). The EC_50_ values of Et_2_O and EtOAc extracts of t037 were 44.3 ± 0.481 µg/mL and 13.5 ± 5.35 µg/mL, respectively. The EC_50_ values of Et_2_O and EtOAc extracts of t100 were 38.0 ± 14.3 µg/mL and 39.1 ± 5.13 µg/mL, respectively. All of them showed high inhibitory activity (Figure 3C). On the other hand, the activity of the hexane and water fractions was relatively low (Figure 3C), suggesting that the aggregation inhibitors have properties intermediate between hydrophobic and hydrophilic. These results indicate that the t037 and t100 extracts can be further partitioned.

### 3.4. Evaluation of the Inhibitory Activity of the EtOAc Extract of t037

The t037 and t100 extracts showed high inhibitory activities in Et_2_O and EtOAc solvents, and the EtOAc extract of t037 had the smallest EC_50_ value among the 10 samples. Therefore, the t037 EtOAc extract was selected for further evaluation. The t037 EtOAc extract was separated by silica gel column fractionation to obtain five fractions (f1–f5). MSHTS analysis showed that f5 had the highest inhibitory activity with an EC_50_ of 2.30 ± 0.859 µg/mL. f5 was further separated by silica gel column fractionation to obtain four fractions (f5f1–f5f4). MSHTS analysis found that f5f3 had the highest inhibitory activity with an EC_50_ of 7.26 ± 6.24 µg/mL. However, compared with f5, f5f3’s inhibitory activity was reduced (Figure 4A,C,D). A comparison of the fluorescence microscopy images clearly demonstrated that f5 inhibited Aβ aggregation more than f5f3 (Figure 4B). To further verify the inhibitory activity of f5 and f5f3, a ThT assay was also performed on them (Appendix A). Compared with the Aβ group only, f5 and f5f3 reduced the fluorescence intensity in a concentration-dependent manner, indicating that f5 and f5f3 have high inhibitory activity against Aβ aggregation toxicity. Whereas f5 inhibited aggregation at 0.2 µg/mL, f5f3 showed no activity unless added at 2 µg/mL. This result also suggests that f5 has a higher aggregation inhibitory activity than f5f3, similar to the result of MSHTS (Figure 4A). These results suggest that some of the compounds of f5 act synergistically through currently unknown mechanisms.

### 3.5. Effect of the t037 EtOAc Extract on Extracellular Aβ Deposition in SH-SY5Y Cells

Previously, we were able to image Aβ aggregation around PC12 and NG108-15 cells with QD nanoprobes [27]. In this study, we analyzed the inhibitory effects of f5 and f5f3 on extracellular Aβ deposition in SH-SY5Y cells, human neuroblastoma cells, to evaluate the effect on human neurons. The results show that the cellular conditions of the Aβ group were similar to those of the f5 and f5f3 treatment groups at different concentrations under bright field observation (Figure 5A). QD fluorescence imaging showed that extracellular Aβ aggregation of the f5 and f5f3 treatment groups at different concentrations was significantly lower than that of the Aβ group (Figure 5A). Through the merged image of Aβ and f5 20 μg/mL, that is, the magnified image combining the bright field and QD field (Appendix A), it can be clearly seen that the cell conditions in the two images are similar, but the fluorescence intensity around the image of f5 20 μg/mL is low, indicating its cell surface deposition inhibition. Moreover, both f5 and f5f3 reduced the QD field’s mean gray value of SH-SY5Y cells (Figure 5B). These results indicate that both f5 and f5f3 effectively inhibited the deposition of extracellular Aβ in SH-SY5Y cells and that the inhibitory effect of f5f3 was similar to that of f5.

## 4. Discussion

In this study, the EC_50_ values of 210 mushrooms were screened, and approximately 5% of them (i.e., 11 species) were found to have high Aβ_42_ aggregation inhibitory activity. Next, we performed an MTT assay on PC12 cells with active mushroom extracts. The results showed that six mushroom crude extracts, including t037 and t100, inhibited Aβ-induced cytotoxicity in neuronally differentiated PC12 cells (Figure 2). PC12 cell toxicity caused by Aβ aggregation was evaluated by another group of researchers using a mixed mushroom mycelia extract (*Phellinus linteus*, *Ganoderma lucidum*, and *Inonotus obliquus*), which effectively inhibited Aβ aggregation and alleviated PC12 cell death [30]. This suggests that our mushroom extract may have similar components to the mushroom mycelia extract. This also verifies that mushrooms have components that inhibit Aβ aggregation toxicity.

Next, based on the EC_50_ results, the EtOAc extract of t037 was selected and further purified through silica gel column chromatography and thin layer chromatography to evaluate the Aβ_42_ inhibitory activity of the extract. After the first separation by silica gel column chromatography, f5 had the best inhibitory activity, which was 10 times higher than the crude extract (Figure 4A). We further separated the compounds with high inhibitory activity in f5 using a second separation step by silica gel column chromatography, but the inhibitory activity of the extracts that were separated decreased (Figure 4A). To verify the accuracy of this result, ThT was used to verify the Aβ inhibitory activities of the two extracts, f5 and f5f3, and the results were consistent with the MSHTS results (Appendix A). This shows that f5 has multiple active substances with Aβ inhibitory activity, and there is a synergistic effect between them. Therefore, the Aβ inhibitory activity of f5 was higher than that of f5f3.

Currently, mushroom compounds with anti-AD potential are divided into several categories, including polysaccharides, steroid compounds, triterpenes, alkaloids, and phenols [31,32,33,34]. Their mechanisms mainly include inhibiting Aβ formation, regulating the NF-κB/MAPK signaling pathway, inhibiting neuronal apoptosis, regulating the immune system, inhibiting acetylcholinesterase, and others [35]. Therefore, based on our analyses, we speculate that the biological activities contained in f5 may belong to several of these categories. Fraction f5 significantly inhibited Aβ aggregation on the surface of SH-SY5Y cells even at a low concentration of 0.16 µg/mL (Figure 5). Since Aβ aggregation and deposition in cells trigger the onset of AD, inhibiting this process may be useful for treating and preventing the disease. Whether there are new components in the f5 fraction or how the components work synergistically is still unknown.

In this study, the inhibitory activity of 210 mushrooms collected in Hokkaido was evaluated against Aβ aggregation by using an automated MSHTS system. We developed this system many years ago to label amyloid proteins with QD fluorescent probes, successfully elucidated the aggregation process of Aβ, tau, α-syn, and other proteins by recording images with real-time fluorescence microscopy, and depicted 3D models of aggregates in 3D images and to screen inhibitors [36]. The detection plate used in this system is a 1536-well plate, and each sample well only needs 5 μL to complete detection, which truly achieves high-throughput, fast, convenient, and accurate screening. In this study, as well as in actual screening results reported in our previous reports, the system can accurately and efficiently evaluate the inhibitory activity, even of crude extracts derived from natural products (plants, mushrooms), as well as various commercial dressings [22,26,37,38,39,40].

In summary, we believe that the MSHTS system will become an important method for detecting amyloid aggregation and screening amyloid aggregation inhibitors. We will explore the ability of this system to screen amyloid inhibitors to develop therapeutic drugs for amyloid diseases and functional foods for the prevention of AD. The EC_50_ of RA is 10.7 µg/mL [26], a well-known Aβ aggregation inhibitor. In contrast, f5 fraction with a higher activity of 2.30 µg/mL was obtained. However, the activity decreased with further purification, so there may be a synergistic effect of multiple compounds that have not yet been elucidated. Elucidating the mechanism of this synergistic effect is one of our future research objectives. The aim of this study was to find a highly active extract. To develop functional foods based on these results, it is necessary to investigate the safety and stability of the f5 fraction, as well as the method of administration in animal models, considering bioavailability and pharmacokinetics. It is also necessary to understand the interactions between the compounds in the fractions and the mechanisms of action on the organism, which will be the subject of the next phase of research.

## 5. Conclusions

We successfully evaluated the Aβ aggregation inhibitory activity of 210 crude extracts of Hokkaido mushrooms using an automated MSHTS system. Among them, 11 mushroom extracts had Aβ inhibitory activity, while 6 of them had neuroprotective effects and significantly increased PC12 cell viability. After solvent separation and purification of two extracts, t037 and t100, the obtained extracts still had good inhibitory activity and neuroprotective properties. In conclusion, our automated MSHTS system has great potential for screening AD inhibitors. In addition to their traditional edible and medicinal values, mushrooms also have potential research value in preventing AD. The Aβ inhibitory activity of f5 was higher than that of other mushrooms. In the future, we would like to conduct in vivo experiments on f5. We hope that f5 may be used as a dietary supplement and Aβ inhibitor for the prevention of AD after multiple experiments.

## Figures and Tables

**Figure 1 foods-13-03740-f001:**
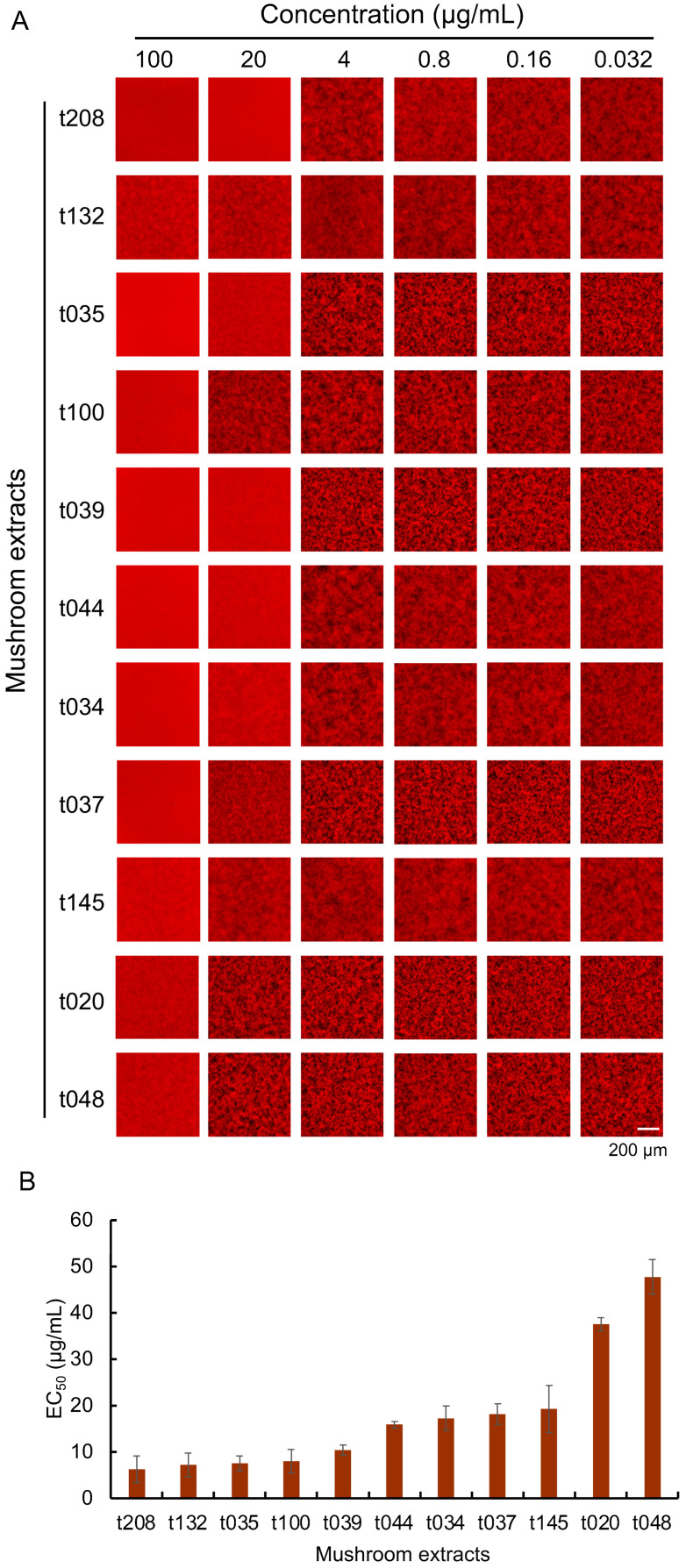
Fluorescence images and EC_50_ values demonstrating the inhibitory effects of mushroom extracts on Aβ aggregation. (**A**) Analysis of Aβ aggregation inhibitory activity of 11 mushroom extracts. Mixed solutions of 25 μM Aβ_42_ and mushroom extracts were added to 1536-well plates and imaged by fluorescence microscopy after incubation at 37 °C for 24 h. The captured images were cropped to 432 × 432 pixels. (**B**) Measurement of EC_50_ values of 11 mushroom extracts. Data are represented as mean ± SD (*n* = 4). Smaller EC_50_ values indicate higher inhibitory activity. EC_50_ values were calculated using Prism GraphPad software.

**Figure 2 foods-13-03740-f002:**
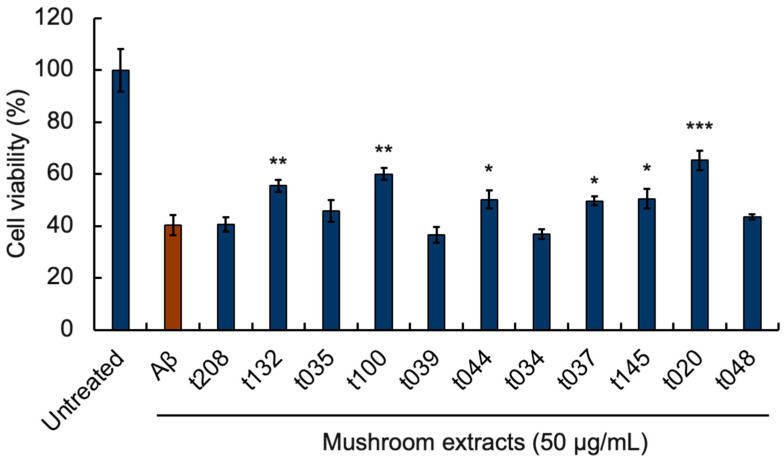
Inhibitory effects of 11 mushroom extracts on Aβ-induced PC12 cell toxicity. PC12 cells were differentiated using NGF for 24 h and treated with 25 μM Aβ_42_ containing each mushroom extract for 24 h. Cell viability measured by the MTT assay is shown as the relative percentage of absorbance of treated samples compared to the control without Aβ and mushroom extract. Each extract was compared with Aβ_42_. Each plot and bar graph represents the mean ± SD (*n* = 3 separate experiments with extracts, six separate experiments in 25 μM Aβ only). (*: *p* < 0.05, **: *p* < 0.01, ***: *p* < 0.001, Welch’s *t*-test).

**Figure 3 foods-13-03740-f003:**
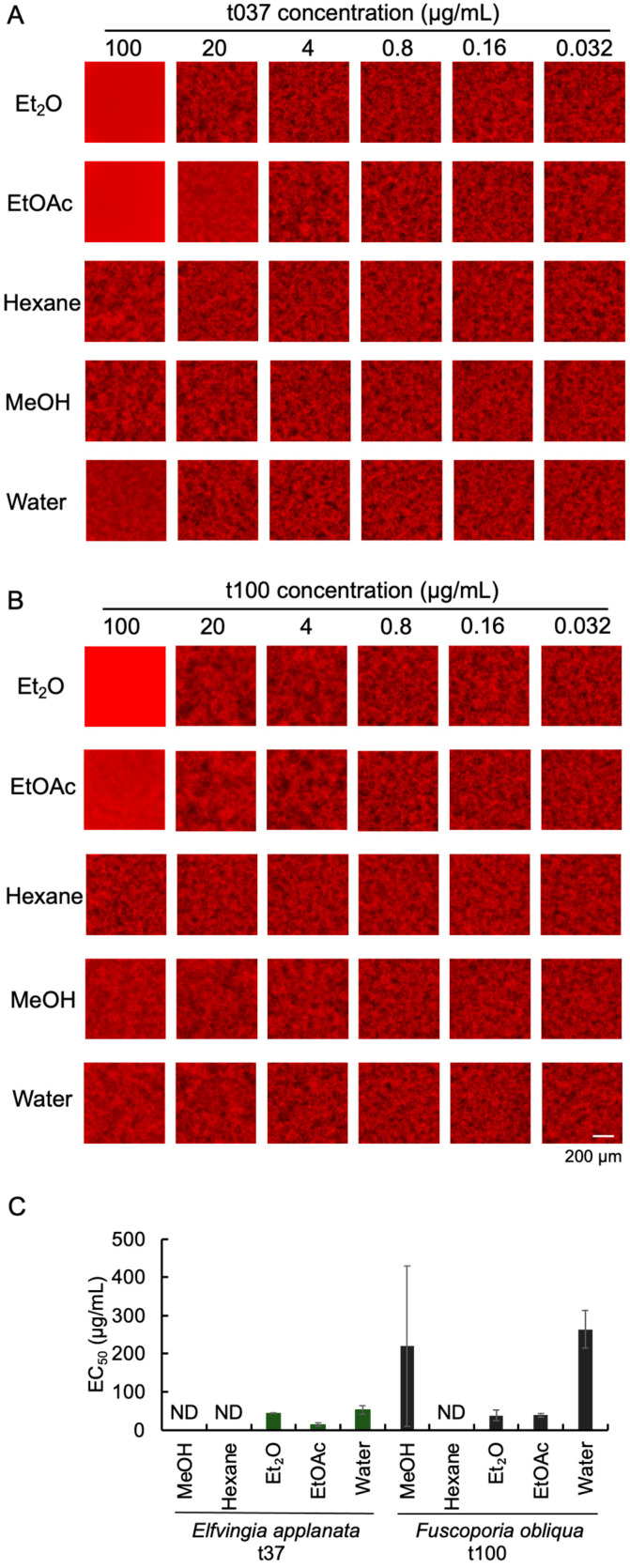
Fluorescence images and EC_50_ values of the inhibitory effect of extracts of 2 mushrooms with different solvents on Aβ aggregation. (**A**) Analysis of the inhibitory activity of t037 at different concentrations on Aβ aggregation in five extraction solvents. (**B**) Analysis of the inhibitory activity of t100 at different concentrations on Aβ aggregation in five extraction solvents. The fluorescence images were cropped to 432 × 432 pixels. (**C**) Measurement of EC_50_ values of t037 and t100 in five extraction solvents. Samples with SD values not less than 50% were considered as unable to calculate EC_50_ values and expressed as not determined (ND). Data are represented as the mean ± SD (*n* = 3).

**Figure 4 foods-13-03740-f004:**
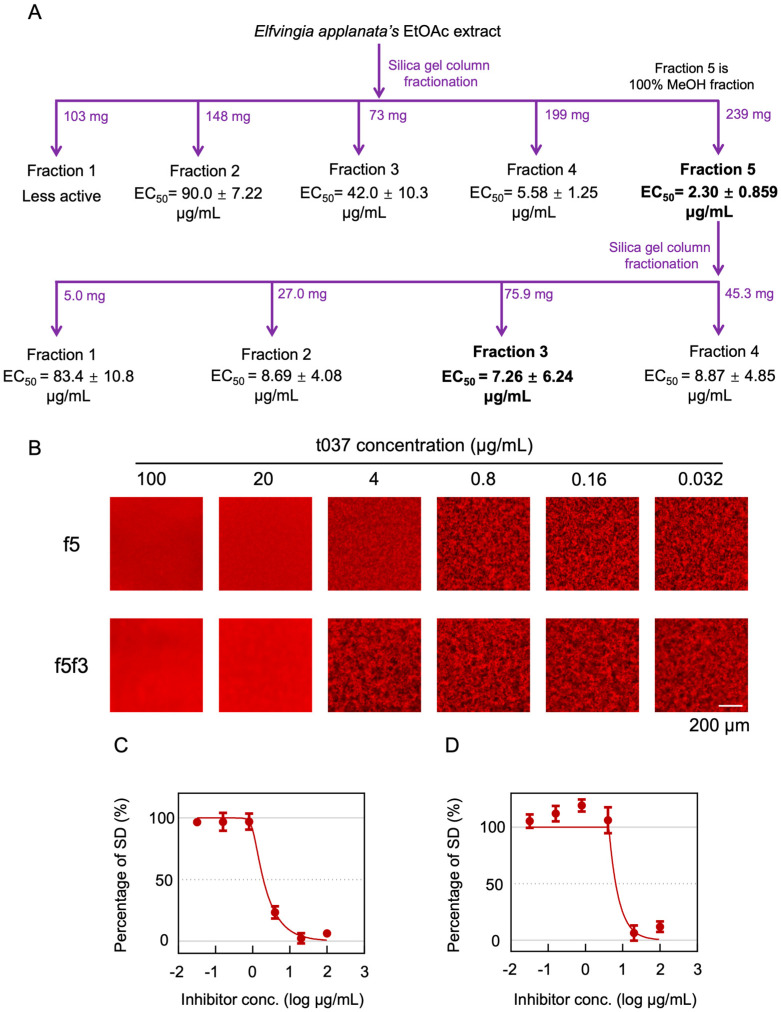
Evaluation of the inhibitory effects of active ingredients in the t037 EtOAc extract. (**A**) The t037 EtOAc extract was separated into five fractions (f1−f5) by silica gel column fractionation. f5 was further separated into four fractions (f5f1−f5f4) by same silica gel column fractionation. (**B**) Fluorescence imaging of the two fractions (f5 and f5f3) with high Aβ aggregation inhibitory activity. The fluorescence images were cropped to 432 × 432 pixels. (**C**) The inhibition curve of f5. (**D**) The inhibition curve of f5f3. The inhibition curves were drawn from the SD values, and EC_50_ was calculated using Prism GraphPad software.

**Figure 5 foods-13-03740-f005:**
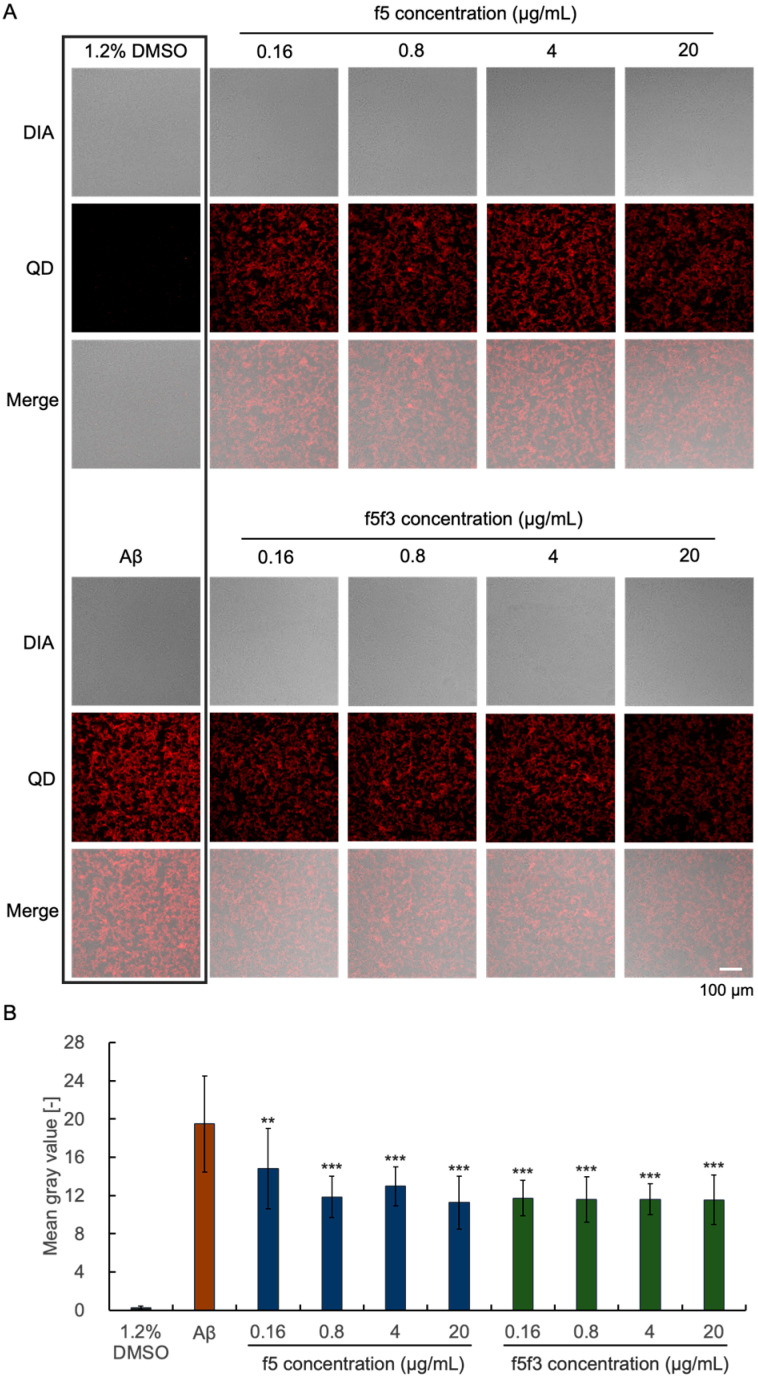
Inhibitory effect of the EtOAc extract of t037 on extracellular Aβ deposition in SH-SY5Y cells. (**A**) SH-SY5Y cells were co-cultured with 25 μM Aβ and 25 nM QDAβ and different concentrations of f5 and f5f3. Images were collected using an inverted fluorescence microscope. 1.2% DMSO was used as the negative control (without Aβ). DIA: bright field image; QD: fluorescence image of QD and Aβ aggregation; Merge: merged image. (**B**) The QD field’s mean gray value of each group was determined. Data are represented as mean ± SD (*n* = 3). (**: *p* < 0.01, ***: *p* < 0.001, Welch’s *t*-test).

**Table 1 foods-13-03740-t001:** Aβ aggregation inhibitory activity and cell viability of 11 mushroom extracts.

Activity Rank	Japanese Name	Scientific Name	EC_50_ (μg/mL)	Cell Viability (%)	Number
1	コフキサルノコシカケ (購入)	*Elfvingia applanata* (purchased)	6.21	40.5	t208
2	コツブタケ	*Pisolithus tinctorius*	7.15	55.4	t132
3	キコブタケ	*Phellinus igniarius*	7.54	45.8	t035
4	カバノアナタケ	*Fuscoporia obliqua*	7.94	60.0	t100
5	サジタケ	*Onnia scaura*	10.4	36.5	t039
6	ハナガサタケ	*Pholiota flammans*	15.8	50.2	t044
7	ツリガネタケ	*Fomes fomentarius*	17.2	36.9	t034
8	コフキサルノコシカケ	*Elfvingia applanata*	18.1	49.6	t037
9	オシロイタケ	*Oligoporus tephroleucus*	19.2	50.5	t145
10	クロカワ	*Boletopsis leucomelas*	37.5	65.3	t020
11	ヌメリスギタケモドキ	*Pholiota aurivella*	47.7	43.4	t048

## Data Availability

The original contributions presented in the study are included in the article/Appendix A, further inquiries can be directed to the corresponding author.

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
