# Peer review of "Screening of a Fraction with Higher Amyloid β Aggregation Inhibitory Activity from a Library Containing 210 Mushroom Extracts Using a Microliter-Scale High-Throughput Screening System with Quantum Dot Imaging"

_foods, 2024, doi:10.3390/foods13233740_

Round 1

Reviewer 1 Report

Comments and Suggestions for Authors

The authors have evaluated the inhibitory effects of various mushroom extracts on Aβ aggregation using PC12 and SH-SY5Y cell models, along with the MSHTS system, providing new insights into the prevention and treatment of Alzheimer's disease. The study is well-structured, and the methodology appears sound. However, some sections require further clarification and revisions.

Comment 1: Given that Section 3.1 evaluates Aβ aggregation inhibition and Section 3.2 assesses neuroprotection in PC12 cells, could differences in these models (Aβ aggregation vs. cell viability) explain the varying results? Might certain compounds show high Aβ inhibition but reduced neuroprotection due to the complex cellular environment or concentration differences?

Comment 2: Why is the threshold of 0.02 mg/mL used to define high Aβ aggregation inhibitory activity? Would it be useful to provide more context or comparison with other known inhibitors to justify this classification?

Comment 3: Are the in vitro experiments using PC12 and SH-SY5Y cell models truly sufficient to evaluate the inhibitory effects of mushroom extracts on Aβ aggregation? What about bioavailability and pharmacokinetics? It would be beneficial to validate these findings using Alzheimer's disease animal models.

Comment 4: Apart from Aβ inhibition activity, what other factors (such as stability or compound interactions) should be considered when developing f5 as a dietary supplement or functional food for the prevention of Alzheimer’s disease?

Comment 5: In certain sections, especially in the Introduction and Discussion, some sentences are quite long and could be split to enhance clarity and comprehension.

Comment 6: P5, line 198, Aβ aggregation was …

Author Response

Comments and Suggestions for Authors:

The authors have evaluated the inhibitory effects of various mushroom extracts on Aβ aggregation using PC12 and SH-SY5Y cell models, along with the MSHTS system, providing new insights into the prevention and treatment of Alzheimer's disease. The study is well-structured, and the methodology appears sound. However, some sections require further clarification and revisions.

Response:

Thank you for your positive comments. We have carefully considered your feedback and made the necessary corrections.

Comment 1:

  1. Given that Section 3.1 evaluates Aβ aggregation inhibition and Section 3.2 assesses neuroprotection in PC12 cells, could differences in these models (Aβ aggregation vs. cell viability) explain the varying results? Might certain compounds show high Aβ inhibition but reduced neuroprotection due to the complex cellular environment or concentration differences?

Response 1:

As the reviewer pointed out, Aβ aggregation inhibitory activity did not necessarily correlate with cell viability. This is probably because some mushroom extracts contain cytotoxic substances. One future goal of this research is the development of functional foods. Therefore, we focused on mushroom extracts that have both Aβ aggregation inhibitory activity and cell viability enhancing activity. We have added these explanations to lines 255-259 of the manuscript.

Comment 2:

  1. Why is the threshold of 0.02 mg/mL used to define high Aβ aggregation inhibitory activity? Would it be useful to provide more context or comparison with other known inhibitors to justify this classification?

Response 2:

We apologize for writing 0.02 mg/mL in the manuscript. In fact, in our previous paper [26], we used values ​​below 0.05 mg/mL as the standard for high activity. We have added an explanation to the manuscript on lines 230-235, including why 0.05 mg/mL was chosen as the threshold. In addition, because mg/mL and µg/mL were mixed in this manuscript, we have unified all of them to µg/mL in the revised manuscript to avoid confusion for readers.

Comment 3:

  1. Are the in vitro experiments using PC12 and SH-SY5Y cell models truly sufficient to evaluate the inhibitory effects of mushroom extracts on Aβ aggregation? What about bioavailability and pharmacokinetics? It would be beneficial to validate these findings using Alzheimer's disease animal models.

Response 3:

Thank you for pointing this out. Our future work is to determine how to efficiently administer the highly active fractions to Alzheimer's disease model animals. We added the explanation in lines 402-407 of the manuscript.

Comment 4:

  1. Apart from Aβ inhibition activity, what other factors (such as stability or compound interactions) should be considered when developing f5 as a dietary supplement or functional food for the prevention of Alzheimer’s disease?

Response 4:

In addition to the discussion related to comment 3, we have also included a discussion of comment 4 in lines 402-407 of the manuscript.

Comment 5:

  1. In certain sections, especially in the Introduction and Discussion, some sentences are quite long and could be split to enhance clarity and comprehension.

Response 5:

According to the comment, we have separated some sentences in the Introduction and Discussion sections, hoping to improve clarity and understanding.

Comment 6:

  1. P5, line 198, Aβ aggregation was …

Response 6:

Thank you for pointing this out. We have changed 'were' to 'was' in line 229-230.

Reviewer 2 Report

Comments and Suggestions for Authors

The study used microliter-scale high-throughput screening (MSHTS) and quantum dot imaging to screen 210 mushroom extracts for compounds inhibiting amyloid-beta (Aβ) aggregation, a key feature of Alzheimer's disease (AD),hence, substances capable of inhibiting this aggregation may be useful for the prevention and treatment of AD. Eleven potent samples were found in Hokkaido, Japan, with Elfvingia applanata and Fuscoporia obliqua extracts showing neuroprotection against Aβ toxicity and were chosen for further analysis. The EtOAc extract of E. applanata showed high Aβ aggregation inhibition, with an EC50 of 0.00230 mg/mL, surpassing rosmarinic acid's 0.0107 mg/mL. Silica gel column chromatography identified the most active fraction (f5) from E. applanata, which also reduced Aβ aggregate deposition on SH-SY5Y cells. The study suggests that f5 could be an AD preventive inhibitor, pending further research. Some major repairs are required.

Q1: The article mentioned 210 mushroom samples collected from Hokkaido. The authors should further clarify whether the samples are representative of all mushroom species and whether they cover enough diversity to support the generalisability of the study's conclusions

Q2: The authors should ensure that the introduction section includes the most recent relevant studies to demonstrate the current progress in the field and the contribution of this manuscript (Nano Lett., 2024, 24, 10418−10425).

Q3: The authors are asked to discuss the potential clinical relevance of the f5 fraction and how it could be translated into actual treatment methods or prevention strategies.

Q4: The authors should provide more information on the identification of active components within the f5 fraction and the specific mechanisms by which these components inhibit Aβ aggregation.

Q5:Ensure that all figures, illustrations, and images have sufficient resolution for clarity upon publication. Additionally, the information in figures and illustrations should be clear and understandable.

Comments on the Quality of English Language

 The English could be improved to more clearly express the research.

Author Response

Comments and Suggestions for Authors:

The study used microliter-scale high-throughput screening (MSHTS) and quantum dot imaging to screen 210 mushroom extracts for compounds inhibiting amyloid-beta (Aβ) aggregation, a key feature of Alzheimer's disease (AD),hence, substances capable of inhibiting this aggregation may be useful for the prevention and treatment of AD. Eleven potent samples were found in Hokkaido, Japan, with Elfvingia applanata and Fuscoporia obliqua extracts showing neuroprotection against Aβ toxicity and were chosen for further analysis. The EtOAc extract of E. applanata showed high Aβ aggregation inhibition, with an EC50 of 0.00230 mg/mL, surpassing rosmarinic acid's 0.0107 mg/mL. Silica gel column chromatography identified the most active fraction (f5) from E. applanata, which also reduced Aβ aggregate deposition on SH-SY5Y cells. The study suggests that f5 could be an AD preventive inhibitor, pending further research. Some major repairs are required.

Response:

Thank you very much for your feedback. We have responded to your comments and suggestions below.

Comment 1:

  1. The article mentioned 210 mushroom samples collected from Hokkaido. The authors should further clarify whether the samples are representative of all mushroom species and whether they cover enough diversity to support the generalisability of the study's conclusions.

Response 1: There are about 1,000 named species of mushrooms in Hokkaido, the mushrooms used in this experiment cover many species and families, but slime molds and cordyceps sinensis have been excluded. These 210 mushrooms include 2 phylum, 6 classes, and 17 orders. We added the explanation in lines 101-104 of the manuscript.

Comment 2:

  1. The authors should ensure that the introduction section includes the most recent relevant studies to demonstrate the current progress in the field and the contribution of this manuscript (Nano Lett., 2024, 24, 1041810425).

Response 2:

To explain the current progress in this field, we have added a QD-related section in the introduction (lines 90-99) that includes the latest QD-related research.

Comment 3:

  1. The authors are asked to discuss the potential clinical relevance of the f5 fraction and how it could be translated into actual treatment methods or prevention strategies.

Response 3:

We added the explanation of potential clinical relevance in lines 402-407 of the manuscript.

Comment 4:

  1. The authors should provide more information on the identification of active components within the f5 fraction and the specific mechanisms by which these components inhibit Aβ aggregation.

Response 4:

We added the explanation about active component in f5 fraction in lines 377-381 of the manuscript.

Comment 5:

Ensure that all figures, illustrations, and images have sufficient resolution for clarity upon publication. Additionally, the information in figures and illustrations should be clear and understandable.

Response 5:

Thank you for pointing this out. We have ensured that all figures, illustrations, and images meet high-resolution standards for optimal clarity in both print and digital formats. All elements within the figures, including labels and legends, have been formatted for readability at standard publication sizes. Additionally, we have structured each figure to present information clearly and concisely. We are happy to make any further adjustments if needed.

Reviewer 3 Report

Comments and Suggestions for Authors

Gegentuya Huanood et al. present a comprehensive study on high-throughput screening of mushroom extracts using quantum-dot hybrids. The study seems comprehensive and scientifically sound but a few deficiencies regard the description of the used molecular QD system.

line 35-90 ff: Introduction requires a paragraph on functionalized colloidal semiconductor quantum dots especially adaptive to biological enviroments. Why are they used and why not conventional dyes? Which hybrid systems are there? How are the QDs stabilized in water? What are the intrinsic fluorescence properties? What other approaches for QD systems could be there (e.g. latest research on QD glutathion systems) Reference to lit. [23] is not sufficient, the description should be self-standing.

line 203 ff: Caption of Fig 1 should clarify what it shows, i.e. "fluorescence images of..." and also which physical parameters are decoded by the intensity maps, i.e. by the red hue and the (density of) black speckles.

line 272 ff. vide surpra (line 203)

line 291 ff. vide supra (line 203)

Author Response

Comments and Suggestions for Authors:

Gegentuya Huanood et al. present a comprehensive study on high-throughput screening of mushroom extracts using quantum-dot hybrids. The study seems comprehensive and scientifically sound but a few deficiencies regard the description of the used molecular QD system.

Response:

Thank you for your positive evaluation. We have carefully considered your comments and made revisions where necessary.

Comment 1:

  1. line 35-90 ff: Introduction requires a paragraph on functionalized colloidal semiconductor quantum dots especially adaptive to biological enviroments. Why are they used and why not conventional dyes? Which hybrid systems are there? How are the QDs stabilized in water? What are the intrinsic fluorescence properties? What other approaches for QD systems could be there (e.g. latest research on QD glutathion systems) Reference to lit. [23] is not sufficient, the description should be self-standing.

Response 1:

Thank you for your comment. We added the QD section explanation in lines 90-99 of the manuscript.

Comment 2:

  1. line 203 ff: Caption of Fig 1 should clarify what it shows, i.e. "fluorescence images of..." and also which physical parameters are decoded by the intensity maps, i.e. by the red hue and the (density of) black speckles.

line 272 ff. vide surpra (line 203)

line 291 ff. vide supra (line 203)

Response 2:

Thank you for your comment. We changed the title of Figure 1 in lines 237-238 of the manuscript.

Reviewer 4 Report

Comments and Suggestions for Authors

Huanood and co-authors prepared a manuscript titled, “Screening of a fraction with higher amyloid baggregation inhibitory activity from a library containing 210 mushroom extracts using a microliter-scale high-throughput screening system with quantum dot imaging,” where the authors describe the application of their MSHTS assay to suggest extracts from two mushrooms with potential utility to reduce amyloid b (Ab) aggregation. However, the manuscript has several critical flaws including missing data and controls, incomplete analyses, and poor presentation as well as discussion of results. As such, I cannot recommend the publication of the manuscript in its current forms. 

Detailed comments are below: 

1)        Background is overly simplified and relevant background is not presented. The authors should (i) provide a brief description of Alzheimer’s disease (e.g., what are the symptoms, how do the biochemical features such as Ab affect pathology); (ii) clearly delineate current drugs used to reduce Ab aggregation and how they compare with mushroom-derived compounds (instead of listing arbitrary nutritional advantages of mushrooms); (iii) define any other neurological benefits of mushroom-derived compounds; (iv) provide a brief description of the known mechanism by which mushroom-derived reduce Ab aggregation propensity; (v) describe the thioflavin T (ThT) assay and how ThT is used to determine Ab aggregation. 

2)        The manuscript lacks rationale for the methodologies used and a description of how they compare to established methodologies. The authors briefly state they use their own assay (MSHTS) for experiments, but they should (i) explain why the MSHTS assay is superior to other assays as well as any limitations; (ii) what are the other alternative methods available to evaluate Abaggregation and why they did not choose to use those methods in this study. 

3)        The goal of the manuscript and the data presented are not correlated. The goal of the manuscript appears to be to test 210 mushrooms to evaluate their ability to prevent Ab aggregation. However, data from only 11 mushrooms are presented and “full” analysis is done for only two mushrooms (which are also not the best choices as discussed later). If the MSHTS method is so superior and the point is to use it to evaluate 210 mushrooms, then the full data set should be presented in the manuscript and/or in the supplement, which currently only holds a table listing the mushroom names, but no data is presented for all 210 mushrooms. Furthermore, the data presented is insufficient to conclude that the mushroom extracts are infact suitable to prevent Ab aggregation. 

4)        The methods should clearly describe the protocols used by the authors. For example, in 2.4, the authors state that mushrooms were harvested for use, however later the authors claim some mushrooms were purchased. Please ensure consistency. Furthermore, the authors claim that mushrooms were chopped after harvesting – however, this could cause contamination of the extracts thereafter with soil and other debris collected during harvesting. Likely, the authors washed and dried the mushrooms – please include this description in the methods. Furthermore, given the utilization of mushroom powders in the extraction procedures, please also list the % water or % moisture remaining in the mushrooms prior to powderization. For mushrooms that were harvested, “from the mountain”, please provide a description of the precise location as well as growth conditions. 

5)        For all methods, please provide the appropriate references if the protocols are based on established protocols, otherwise, please provide a detailed rationale for the protocol used. 

a.        For example, in 2.5, what media was the growth factor added into? The culture and differentiation of PC12 cells is widely established and the authors should be using standard reported procedures, but no references were cited. 

b.        As another example, in 2.6, the authors use MeOH, hexane, Et2O, EtOAc, and chloroform extraction. All of these solvents are toxic for consumption and yet the end goal of the extracted mushroom compounds is for human consumption to treat Alzheimer’s disease. So why did the authors use these solvents for extraction? If the choice is based on prior literature, please provide the appropriate references. 

c.        Furthermore, the authors fractionate the elutions – please provide fractionation volumes for each fractionation and sub-fractionation performed and explain how each “fraction” was determined. 

d.        As another example, in 2.7, please explain the ThT assay and why 24 h was used as the incubation timepoint. 

e.        As another example, please provide the appropriate references for 2.8 that show mean gray value can represent Ab aggregation. In general, brightfield imaging highlights all heterogeneities in cells and the gray value can be affected by the presence of lysosomes, intracellular vesicles, aggregations or clumping of cells, blebbing, etc. 

6)        If the authors claim to have evaluated 210 mushrooms, the MSHTS data for all 210 mushrooms must be shown in the supplement, the data must be quantified and a clear rationale should be presented for choosing the 11 mushrooms shown in the main text.

7)        The authors must present a clear rationale for their choice of mushrooms for subsequent characterization.

a.        For example, from Fig 1, t208, t039, t044 seem to have superior performance to t100, t037, which were chosen by the authors because they suppress aggregation to a higher extent at lower concentrations. Even from the EC50 values, it is apparent that t208 and t132 have comparable performance to t100 and superior performance to t037. So, the authors’ choice is again unclear and at the very least the authors should repeat the later experiments with t208 and t132. 

b.        From the PC12 cell results in Fig 2, the top choices should be t020 and t100 (see also Table 1), while t208, t039, t044 had poor or median viabilities. This brings to question (again) why the authors chose t037 as well as how the MSHTS results correlate with the PC12 cell viabilities, which in fact look almost completely opposite to the MSHTS results. These clearly polar results mean that all 11 extracts should move on to the third round of testing. Otherwise, the authors should provide clear rationale for the two sets of data presented in Fig1 and 2 and explain why the results are at all meaningful. 

c.        Moreover, the authors should check that the row of t132 in Fig 1 is not flipped, because it is odd that there is less aggregation at 0.032 ug/mL than at 100 ug/mL and should also check the images for t208, t132, t044, t034, t145, which do not appear to be in focus.

8)        The authors should quantify the Ab aggregate sizes from their MSHTS assay, as shown in Figure 1. This would provide a clear numerical metric by which to determine the most effective mushroom at preventing Ab aggregation.  

9)        The authors should evaluate the time-dependent effects of their mushroom-based extracts on Ab aggregation. The MSHTS and EC50 is only tested at 24h, however, at the highest concentration (100ug/mL) where there is no apparent difference between the compounds, a kinetic analysis at multiple time points up to 24h may provide interesting and new insight into the mechanistic efficiency of mushroom-based treatments. In this line, the authors should also clearly report in the figure caption what the concentration of extract used was for the EC50 tests. 

10)  The authors should compare the MSHTS and EC50 on state-of-the-art drugs and other known mushroom-derived compounds as a control.

11)  The PC12 cell viability data is insufficient to conclude efficiency of the mushroom-derived compounds utility in reducing Ab aggregation.

a.        First, the authors need to provide the untreated PC12 baseline for the results (which should be > 90% viability). In general, < 80% viability for cells is very low and thus the efficacy of the data is questionable. 

b.        Please provide relevant references that show that PC12 cell viability is a relevant metric for determining Ab aggregation.

c.        The authors must provide confirmation that the NGF treatment indeed caused differentiation of the PC12 cells and the relative populations of differentiated cells must be quantified for each case (how do we know that the mushroom extracts are not more potent to one cells type versus another?).

d.        Please repeat the experiments with all of the mushroom extracts - Ab to show that the mushroom extracts themselves are not further compromising the viability of the cells. 

e.        Please also repeat the Ab + mushroom extract experiments at multiple time points (e.g., 3,6,12,18, and 24h). 

f.           Please also provide representative fluorescence images for each condition showing the relative viabilities describe in the supplement. 

12)  The authors should clearly explain why different extraction methods are used in Fig 1 and 3 and provide quantification of the aggregate sizes observed with the different extraction methods. The results suggest that the Ab aggregation behavior is different depending on the solvent/extraction method used, if this is the case then the different solvent-based extraction methods should be repeated for all mushrooms and the most effective solvent-mushroom pair should be determined. 

a.        Please also explain the difference in intensity observed for Fig3A EtwO and 3B Et2O at 100 ug/mL. 

b.        Please also explain why in Fig 3 the EC50 for t100 is higher with MeOH and water than with the method used in Fig 1. 

c.        Please also explain why several of the EC50 values in Fig 3 were not determined and provide the concentration of Ab used for EC50 measurements in Fig 1 and 3 in the figure captions. 

13)  The authors must explain which method is most accurate for determining the efficiency of mushroom-derived compounds. The authors present multiple methods across the paper and each presents different results, these results must be tied together and rationally explained. Currently the data presented only suggest that while mushroom-derived extracts may limit Ab aggregation, there is no conclusive experiment that suggests the strongest candidate. As a result, the manuscript, while containing interesting data, is pointless.

a.        Please also explain your rationale for switching to SH-SY5Y cells as opposed to continuing to use PC12 cells for the QD assay. 

14)  For the data presented in Fig 5, please present appropriate controls and higher magnification images. As mentioned previously, the authors need to first prove that the gray value is a reliable metric for Ab aggregation. After which, the author should provide higher magnification DIA images for each condition to prove that the cell morphology and intracellular components are unchanged and that the changes in gray value are coming from Ab aggregation. Furthermore, the authors will then need to show the gray value for images obtained from cells untreated with Ab or DMSO as well as cells treated with the relevant fractions of mushroom extract but without Ab to confirm that the gray values changes are in fact from changes in the relative concentration of Ab aggregation. Moreover, the authors should also explain the relevance of the QD intensity, distribution and profile as they do not quantify QD images and the purpose of the exercise is unclear. Finally, the authors’ conclusions again do not match their data. The authors claim that f5 is more effective at reducing Abaggregation than f5f3, but Fig 5B clearly shows that f5f3 is able to reduce Ab aggregation to a similar level as higher concentrations of f5, suggesting that f5f3 is more effective. The authors need to re-evaluate their data and provide clear explanations for the conclusions they draw. 

15)  The discussion section is a reiteration of the introduction and results and should be completely re-written. The purpose of the discussion section is to discuss the results and should be used in this way. Please re-write with less focus on what you did before and focus on the results that you are obtaining now.

16)  The conclusions are a redundant summary of their results and should be re-written to focus on the key points of their findings with an outlook into future directions of work.

17)  The references provided in the manuscript are both insufficient and, in some cases, questionable. For example, in the introduction, the authors should provide multiple reviews on Alzheimer’s disease pathology and the currently known mechanisms for the pathology. The papers cited should also directly correlate to the main points discussed in the paper. For example, the purpose of reference 13, 26,27,30 are unclear. 

Language: Huanood and co-authors have some issues with language although the manuscript is generally readable, it is often redundant and unnecessarily wordy (e.g., the title). I recommend the authors focus more on conveying the results clearly and using quantitative metrics rather than on attempting to tie their results to disease treatment.

Author Response

Huanood and co-authors prepared a manuscript titled, “Screening of a fraction with higher amyloid b aggregation inhibitory activity from a library containing 210 mushroom extracts using a microliter-scale high-throughput screening system with quantum dot imaging,” where the authors describe the application of their MSHTS assay to suggest extracts from two mushrooms with potential utility to reduce amyloid b (Ab) aggregation. However, the manuscript has several critical flaws including missing data and controls, incomplete analyses, and poor presentation as well as discussion of results. As such, I cannot recommend the publication of the manuscript in its current forms.

Detailed comments are below:

Response:

Thank you for your comments. We have considered your comments carefully and made necessary revisions to the manuscript. We have provided responses to all comments below.

Comment 1:

1) Background is overly simplified and relevant background is not presented. The authors should (i) provide a brief description of Alzheimer’s disease (e.g., what are the symptoms, how do the biochemical features such as Ab affect pathology); (ii) clearly delineate current drugs used to reduce Ab aggregation and how they compare with mushroom-derived compounds (instead of listing arbitrary nutritional advantages of mushrooms); (iii) define any other neurological benefits of mushroom-derived compounds; (iv) provide a brief description of the known mechanism by which mushroom-derived reduce Ab aggregation propensity; (v) describe the thioflavin T (ThT) assay and how ThT is used to determine Ab aggregation.

Response 1:

(i) The symptoms of Alzheimer's disease are briefly described in lines 37-38.

How biochemical features such as Aβ affect pathology is answered in lines 39-44.

(ii) Current drugs used to reduce Aβ aggregation are described in lines 44-54.

(iii)-(iv) Any other neurological benefits of mushroom-derived compounds and known mechanisms by which mushroom-derived compounds related to reduce the Aβ aggregate are answered in lines 68-75.

(v) The principle of the Thioflavin T (ThT) assay and how it relates to Aβ aggregation is described in lines 78-81, and the concentration of ThT is answered in 2.7 of the method.

ThT's shortcomings are in lines 81-83.

Comment 2:

2) The manuscript lacks rationale for the methodologies used and a description of how they compare to established methodologies. The authors briefly state they use their own assay (MSHTS) for experiments, but they should (i) explain why the MSHTS assay is superior to other assays as well as any limitations; (ii) what are the other alternative methods available to evaluate Ab aggregation and why they did not choose to use those methods in this study.

Response 2:

(i) The advantages of the MSHTS system are given in lines 87-88.

(ii) Other methods that can be used to evaluate Aβ aggregation include ThT and TEM.

The disadvantages of ThT are answered in the previous question. The disadvantages of TEM are in lines 83-84.

Comment 3:

3) The goal of the manuscript and the data presented are not correlated. The goal of the manuscript appears to be to test 210 mushrooms to evaluate their ability to prevent Ab aggregation. However, data from only 11 mushrooms are presented and “full” analysis is done for only two mushrooms (which are also not the best choices as discussed later). If the MSHTS method is so superior and the point is to use it to evaluate 210 mushrooms, then the full data set should be presented in the manuscript and/or in the supplement, which currently only holds a table listing the mushroom names, but no data is presented for all 210 mushrooms. Furthermore, the data presented is insufficient to conclude that the mushroom extracts are infact suitable to prevent Ab aggregation.

Response 3:

We put all the EC50 values in the Supplemental Table S1. Mushroom extracts from 210 species were evaluated, and inhibitory activity was detected in extracts from 18 species.

Comment 4:

4) The methods should clearly describe the protocols used by the authors. For example, in 2.4, the authors state that mushrooms were harvested for use, however later the authors claim some mushrooms were purchased. Please ensure consistency. Furthermore, the authors claim that mushrooms were chopped after harvesting – however, this could cause contamination of the extracts thereafter with soil and other debris collected during harvesting. Likely, the authors washed and dried the mushrooms – please include this description in the methods. Furthermore, given the utilization of mushroom powders in the extraction procedures, please also list the % water or % moisture remaining in the mushrooms prior to powderization. For mushrooms that were harvested, “from the mountain”, please provide a description of the precise location as well as growth conditions.

Response 4:

According to reviewer comments, we added information about where to collect mushrooms and a detailed processing method containing extraction conditions to session 2.4.

Comment 5a:

5) For all methods, please provide the appropriate references if the protocols are based on established protocols, otherwise, please provide a detailed rationale for the protocol used.

  1. For example, in 2.5, what media was the growth factor added into? The culture and differentiation of PC12 cells is widely established and the authors should be using standard reported procedures, but no references were cited.

Response 5a:

Regarding this comment, a reference has been added in 2.5 of the method.

Comment 5b:

  1. As another example, in 2.6, the authors use MeOH, hexane, Et2O, EtOAc, and chloroform extraction. All of these solvents are toxic for consumption and yet the end goal of the extracted mushroom compounds is for human consumption to treat Alzheimers disease. So why did the authors use these solvents for extraction? If the choice is based on prior literature, please provide the appropriate references.

Response 5b:

This is a common fractionation method in natural product science to find active fractions, but the fractions separated by solvent are not intended to be administered directly to humans. How to prepare samples for administration to humans is another issue.

Comment 5c:

  1. Furthermore, the authors fractionate the elutions – please provide fractionation volumes for each fractionation and sub-fractionation performed and explain how each “fraction” was determined.

Response 5c:

The volume of each fraction and sub-fractionation was approximately 50 mL, and each fraction was determined by TLC results (lines 183-184).

Comment 5d:

  1. As another example, in 2.7, please explain the ThT assay and why 24 h was used as the incubation timepoint.

Response 5d:

Since Aβ aggregation is known to be fully saturated at 24 h [21, 22], the ThT assay was performed using 24 h as the incubation time point (Line 197).

Comment 5e:

  1. As another example, please provide the appropriate references for 2.8 that show mean gray value can represent Ab aggregation. In general, brightfield imaging highlights all heterogeneities in cells and the gray value can be affected by the presence of lysosomes, intracellular vesicles, aggregations or clumping of cells, blebbing, etc.

Response 5e:

In this paper, we compared the gray values ​​of QD fluorescence colocalized with Aβ, rather than bright-field images, which are influenced by intracellular structures. In fact, in the negative control sample without Aβ (Fig. 5A and 5B, 1.2 % DMSO), the gray value is close to 0. To aid the reader's understanding, we have added information to the legends in Figure 5 (line 343 and 344).

Comment 6:

6) If the authors claim to have evaluated 210 mushrooms, the MSHTS data for all 210 mushrooms must be shown in the supplement, the data must be quantified and a clear rationale should be presented for choosing the 11 mushrooms shown in the main text.

Response 6:

As mentioned in response 3, we added all the EC50 values in the Supplemental Table S1.

Comment 7a:

7) The authors must present a clear rationale for their choice of mushrooms for subsequent characterization.

  1. For example, from Fig 1, t208, t039, t044 seem to have superior performance to t100, t037, which were chosen by the authors because they suppress aggregation to a higher extent at lower concentrations. Even from the EC50 values, it is apparent that t208 and t132 have comparable performance to t100 and superior performance to t037. So, the authors’ choice is again unclear and at the very least the authors should repeat the later experiments with t208 and t132.

Response 7a:

They were selected based on their aggregation inhibitory activity, viability, and also availability (lines 270-272).

Comment 7b:

  1. From the PC12 cell results in Fig 2, the top choices should be t020 and t100 (see also Table 1), while t208, t039, t044 had poor or median viabilities. This brings to question (again) why the authors chose t037 as well as how the MSHTS results correlate with the PC12 cell viabilities, which in fact look almost completely opposite to the MSHTS results. These clearly polar results mean that all 11 extracts should move on to the third round of testing. Otherwise, the authors should provide clear rationale for the two sets of data presented in Fig1 and 2 and explain why the results are at all meaningful.

Response 7b:

We have added a response to this comment on lines 255-259.

Comment 7c:

  1. Moreover, the authors should check that the row of t132 in Fig 1 is not flipped, because it is odd that there is less aggregation at 0.032 ug/mL than at 100 ug/mL and should also check the images for t208, t132, t044, t034, t145, which do not appear to be in focus.

Response 7c:

Thank you for your careful check. I rechecked the fluorescence image of t132 in Fig. 1 and found that the fluorescence image at 0.032 μg/mL was misplaced as you said. Now I put the correct position image. Indeed, the images t208, t132, t044, t034, and t145 appear to be slightly out of focus, but we have validated the automated MSHTS system in a previous paper [26] and do not believe this will affect the results.

Comment 8:

8) The authors should quantify the Ab aggregate sizes from their MSHTS assay, as shown in Figure 1. This would provide a clear numerical metric by which to determine the most effective mushroom at preventing Ab aggregation. 

Response 8:

Since the focus here is on inhibition of aggregation, the effect on aggregate shape and size is another issue.

Comment 9:

9) The authors should evaluate the time-dependent effects of their mushroom-based extracts on Ab aggregation. The MSHTS and EC50 is only tested at 24h, however, at the highest concentration (100ug/mL) where there is no apparent difference between the compounds, a kinetic analysis at multiple time points up to 24h may provide interesting and new insight into the mechanistic efficiency of mushroom-based treatments. In this line, the authors should also clearly report in the figure caption what the concentration of extract used was for the EC50 tests.

Response 9:

The EC50 of MSHTS was only tested at 24 h because our previous reports [21, 22, 26] determined that Aβ aggregation was completely saturated at 24 h. At the highest concentration (100 ug/mL), there were no significant differences between the compounds because all had an inhibitory effect at the highest concentration, most of which appeared as a uniform red fluorescence image (the aggregation state would be represented by black and red fluorescence images). As pointed out by the reviewer, kinetic analysis at multiple time points up to 24 h may provide interesting and new insights into the mechanistic efficiency of mushroom treatment, but this is future work.

All dried extracts and fractions were dissolved in DMSO to a concentration of 10 mg/mL for use (lines 187-188).

Comment 10:

10) The authors should compare the MSHTS and EC50 on state-of-the-art drugs and other known mushroom-derived compounds as a control.

Response 10:

We have not checked the EC50 of the state-of-the-art drugs (such as Aβ monoclonal antibodies) by the MSHTS system. As for other known mushroom-derived compounds, we bought the following 4 compounds, performed the MSHTS, and found that the Aβ aggregation inhibitory activity was not high (Because activities were very low, the results were not presented in this paper). We bought Ganoderic acid A, which is a compound in Ganoderma lucidum (Ganoderma lucidum is very similar to t37 in our paper). We also bought Syringic acid, Caffeic acid and 3,4-Dihydroxybenzoic acid, all compounds in t100.

Comment 11a:

11) The PC12 cell viability data is insufficient to conclude efficiency of the mushroom-derived compounds utility in reducing Ab aggregation.

  1. First, the authors need to provide the untreated PC12 baseline for the results (which should be > 90% viability). In general, < 80% viability for cells is very low and thus the efficacy of the data is questionable.

Response 11a:

A 100% bar was added to the graph in Figure 2 to indicate the presence of untreated samples.

Comment 11b and 11c:

  1. Please provide relevant references that show that PC12 cell viability is a relevant metric for determining Ab aggregation.
  2. The authors must provide confirmation that the NGF treatment indeed caused differentiation of the PC12 cells and the relative populations of differentiated cells must be quantified for each case (how do we know that the mushroom extracts are not more potent to one cells type versus another?).

Response 11b and 11c:

We have answered this in lines 245-247.

Comment 11d:

  1. Please repeat the experiments with all of the mushroom extracts - Ab to show that the mushroom extracts themselves are not further compromising the viability of the cells.

Response 11d:

None of the samples were more cytotoxic than the one containing no inhibitor (Figure 2), indicating that the mushroom extract itself is unlikely to have significant cytotoxicity.

Comment 11e:

  1. Please also repeat the Ab + mushroom extract experiments at multiple time points (e.g., 3,6,12,18, and 24h).

Response 11e:

It is unclear why toxicity needs to be measured over time, this experiment is not necessary at this stage. If it is to clarify the mechanism, it is a future task.

Comment 11f:

  1. Please also provide representative fluorescence images for each condition showing the relative viabilities describe in the supplement.

Response 11f:

The supplement contains ThT assay, which has nothing to do with cell experiments.

Comment 12:

12)  The authors should clearly explain why different extraction methods are used in Fig 1 and 3 and provide quantification of the aggregate sizes observed with the different extraction methods. The results suggest that the Ab aggregation behavior is different depending on the solvent/extraction method used, if this is the case then the different solvent-based extraction methods should be repeated for all mushrooms and the most effective solvent-mushroom pair should be determined.

Response 12:

Fig. 1 shows the evaluation of the extract library to select candidate mushrooms. Because fractionation of crude extracts requires large amounts of sample, the extraction method for fractionation is slightly different from that used for library construction, although the solvent polarity is similar.

Comment 12a:

  1. Please also explain the difference in intensity observed for Fig3A EtwO and 3B Et2O at 100 ug/mL.

Response 12a:

The difference is whether QD nanoprobes are dispersed or aggregated. If it aggregates, light and dark areas will appear. The interpretation of these results is also explained in cited papers, so we do not believe it is necessary to explain it again in this paper.

Comment 12b:

  1. Please also explain why in Fig 3 the EC50 for t100 is higher with MeOH and water than with the method used in Fig 1.

Response 12b:

As mentioned in Response 12, there is a slight difference between samples in libraries and samples that have been fractionated on a large scale.

Comment 12c:

  1. Please also explain why several of the EC50 values in Fig 3 were not determined and provide the concentration of Ab used for EC50 measurements in Fig 1 and 3 in the figure captions.

Response 12c:

The response to this comment is in lines 290-291.

The concentration of Aβ in all experiments was 25μM (line 131).

Comment 13:

13) The authors must explain which method is most accurate for determining the efficiency of mushroom-derived compounds. The authors present multiple methods across the paper and each presents different results, these results must be tied together and rationally explained. Currently the data presented only suggest that while mushroom-derived extracts may limit Ab aggregation, there is no conclusive experiment that suggests the strongest candidate. As a result, the manuscript, while containing interesting data, is pointless.

Response 13:

Because the activity of the hexane and water extract fractions was low, it is speculated that the substance has intermediate properties between hydrophobic and hydrophilic (lines 281-283).

Comment 13a:

  1. Please also explain your rationale for switching to SH-SY5Y cells as opposed to continuing to use PC12 cells for the QD assay.

Response 13a:

We switched to SH-SY5Y cells because we wanted to see the effect on human neurons (lines 326-327).

Comment 14:

14)  For the data presented in Fig 5, please present appropriate controls and higher magnification images. As mentioned previously, the authors need to first prove that the gray value is a reliable metric for Ab aggregation. After which, the author should provide higher magnification DIA images for each condition to prove that the cell morphology and intracellular components are unchanged and that the changes in gray value are coming from Ab aggregation. Furthermore, the authors will then need to show the gray value for images obtained from cells untreated with Ab or DMSO as well as cells treated with the relevant fractions of mushroom extract but without Ab to confirm that the gray values changes are in fact from changes in the relative concentration of Ab aggregation. Moreover, the authors should also explain the relevance of the QD intensity, distribution and profile as they do not quantify QD images and the purpose of the exercise is unclear. Finally, the authors’ conclusions again do not match their data. The authors claim that f5 is more effective at reducing Ab aggregation than f5f3, but Fig 5B clearly shows that f5f3 is able to reduce Ab aggregation to a similar level as higher concentrations of f5, suggesting that f5f3 is more effective. The authors need to re-evaluate their data and provide clear explanations for the conclusions they draw.

Response 14:

We have framed the control group in Figure 5 and added additional information in the legend. We have also placed higher magnification images of Aβ and f5 20μg/mL in Supplementary Figure S2, and the explanation was added in lines 331-335.

The QD field in the Aβ-free sample (1.2% DMSO) is completely dark (Figure 5A and 5B), suggesting that the average gray value of the QD is a reliable indicator of Aβ aggregation. As pointed out by the reviewer, Figure 5B shows that f5f3 appears to be able to reduce Aβ deposition on the cell surface more than f5, but the difference was not significant. On the other hand, the aggregation inhibitory activity differed by more than three-fold (Figure 4).

Comment 15:

15) The discussion section is a reiteration of the introduction and results and should be completely re-written. The purpose of the discussion section is to discuss the results and should be used in this way. Please re-write with less focus on what you did before and focus on the results that you are obtaining now.

Response 15:

We corrected the discussion section to focus on the results we obtained.

Comment 16:

16)  The conclusions are a redundant summary of their results and should be re-written to focus on the key points of their findings with an outlook into future directions of work.

Response 16:

We corrected the conclusion section.

Comment 17:

17) The references provided in the manuscript are both insufficient and, in some cases, questionable. For example, in the introduction, the authors should provide multiple reviews on Alzheimer’s disease pathology and the currently known mechanisms for the pathology. The papers cited should also directly correlate to the main points discussed in the paper. For example, the purpose of reference 13, 26,27,30 are unclear.

Response 17:

Thank you for your comment. The references with unclear purpose and the content in the discussion that is the same as the introduction have been deleted.

Comment 18:

Language: Huanood and co-authors have some issues with language although the manuscript is generally readable, it is often redundant and unnecessarily wordy (e.g., the title). I recommend the authors focus more on conveying the results clearly and using quantitative metrics rather than on attempting to tie their results to disease treatment.

Response 18:

Thank you for your suggestion. We have once again corrected the language issues of this paper.

Round 2

Reviewer 1 Report

Comments and Suggestions for Authors

Consider adding a schematic summarizing the workflow of the MSHTS screening process to visually aid understanding.

Author Response

Thank you for your comment. A schematic summarizing the workflow of the MSHTS screening process is shown in Fig. 6 of the cited paper (ref. 22, Ishigaki et al., 2013). The automated MSHTS workflow is also shown in Fig. 1e of the cited paper (ref. 26, Sasaki et al., 2019). Screening using the MSHTS system has been performed many times (refs. 22, 26, 37, 38, 39, 40), so there is no need to repeat the workflow in this paper.

Reviewer 2 Report

Comments and Suggestions for Authors

The article, in its present form, is much improved with respect to its previous edition. All the questions I raised have been well addressed or explained. I recommend it for publication without further revision.

Comments on the Quality of English Language

no

Author Response

We thank you for your comments which have improved this paper.